# Adjuvant Therapy with Budesonide Post-Kasai Reduces the Need for Liver Transplantation in Biliary Atresia

**DOI:** 10.3390/jcm10245758

**Published:** 2021-12-09

**Authors:** Joachim F. Kuebler, Omid Madadi-Sanjani, Eva D. Pfister, Ulrich Baumann, David Fortmann, Johannes Leonhardt, Benno M. Ure, Michael P. Manns, Richard Taubert, Claus Petersen

**Affiliations:** 1Department of Pediatric Surgery, Hannover Medical School, 30625 Hannover, Germany; kuebler.joachim@mh-hannover.de (J.F.K.); madadi-sanjani.omid@mh-hannover.de (O.M.-S.); fortmann.david@mh-hannover.de (D.F.); ure.benno@mh-hannover.de (B.M.U.); 2Department of Paediatric Gastroenterology and Hepatology, Hannover Medical School, 30625 Hannover, Germany; pfister.eva-doreen@mh-hannover.de (E.D.P.); baumann.u@mh-hannover.de (U.B.); 3Clinic for Pediatric Surgery, Klinikum Braunschweig, 38118 Braunschweig, Germany; j.leonhardt@klinikum-braunschweig.de; 4Department of Gastroenterology, Hepatology and Endocrinology, Hannover Medical School, 30625 Hannover, Germany; manns.michael@mh-hannover.de (M.P.M.); taubert.richard@mh-hannover.de (R.T.)

**Keywords:** biliary atresia, budesonide, adjuvant therapy, liver transplantation

## Abstract

Based on the hypothesis that autoimmunological factors coregulate the pathomechanism in biliary atresia (BA), adjuvant therapy with steroids has become routine, although its efficacy has never been proven. In 2010, a study on the advantages of budesonide compared to prednisolone in autoimmune hepatitis gave rise to experimental therapy using budesonide as an adjuvant BA treatment. Ninety-five BA patients prospectively received a budesonide 2 mg/dose rectal foam daily for three months (SG). A case-matched control group (CG: 81) was retrospectively recruited. The outcome measures were survival with native liver (SNL), determined at six months and two years after the Kasai procedure. The follow-up rate was 100%. At six months, SNL was statistically not different but became so after two years (SG: 54%; CG: 32%; *p* < 0.001). No steroid-related side effects were observed, except for eight patients with finally caught-up growth retardation. This study demonstrates for the first time a significantly longer survival with native liver in patients with BA after adjuvant therapy. However, indication, dosage, and duration of any budesonide application is not given in neonates with BA. Hence, we suggest extending the postoperative use of budesonide in a multicenter observational study with a clearly defined follow-up protocol, particularly in terms of potentially underestimated side effects.

## 1. Introduction

The most frequent indication of pediatric liver transplantation (LTx) is given in patients with biliary atresia (BA) within the course of an unfavorable outcome after the Kasai procedure [1]. Even under the best circumstances and after early referral to pediatric liver units, the overall outcome remains unsatisfying and the survival with native liver (SNL) drops below 30% over the long term [2,3]. As long as the etiology and the pathomechanism of BA are not yet understood, Kasai portoenterostomy (KPE) per se or sequential surgery by KPE and LTx remain the only therapeutic options and provide an overall survival rate of about 90% [1,4].

On the basis of the hypothesis that BA is a triggered inflammatory process, following (auto)immunological patterns [5,6,7], the perioperative administration of corticosteroids has become routine. Despite the fact that this treatment has no beneficial effect on LTx incidence, which has been demonstrated in several well-designed studies, postoperative steroids are still used by most pediatric surgeons [8,9,10].

In 2010, Manns et al. reported that the induction of biochemical remission in patients with autoimmune hepatitis (AIH) would be more effective using budesonide instead of prednisolone, both in combination with azathioprine [11]. Salvage therapy with budesonide was also capable of doubling remission rate in difficult to treat AIH patients [12,13]. Likewise, when added to standard therapy with ursodesoxycholic acid [14,15,16], budesonide improved the therapeutic effects on primary biliary cholangitis (PBC), an autoimmune liver disease manifesting mostly in adults. Although it was a highly speculative idea that the pathomechanism could be in any way comparable in AIH, PBC and BA, this paper gave reason to consider potentially similar effects in biliary atresia. Following a thorough interdisciplinary discussion and detailed consideration, a clinical trial protocol was prepared. From 2011 onward, when the Kasai procedure was scheduled, we offered this experimental adjuvant therapy to parents. After reviewing the first cases, we were already observing an increasing number of patients whose bilirubin turned normal, which encouraged us to pursue this attempt. This trend stabilized during the following years and, from an ethical point of view, we realized that we could not deprive future cases of this option. On the other hand, patients with rare diseases, for whom mid- and long-term observation is inevitable, require a longer follow-up period in order to obtain statistically sound results. Hence, we continued with the protocol of rectally applied budesonide after KPE until passing the number of more than 120 patients. We then reviewed their data in the context of survival with a native liver and the need for liver transplantation at defined and reproducible reference dates.

## 2. Materials and Methods

### 2.1. Diagnostics and Treatment

From May 2011 onward we informed all parents of BA-patients about the option of an experimental treatment with budesonide. During a reflection period lasting until the fifth postoperative day, scheduled for the beginning of the treatment, the team was consistently available for discussing further details. After another extensive medical briefing, the parents then signed their informed consent for the off-label use of budesonide, which included details of the patients’ age, the indication, the dosage, and the application. They were also informed about any potential side effects and assured that they could stop the treatment by tapering the dosage at any time without giving reasons. The parents also agreed that patients would be closely followed-up and that their data would be prospectively registered. A retrospective analysis of this data has been approved by the ethical committee of Hannover Medical School (No. 9429_BO_K_2020).

With being a tertiary referral center, all patients with neonatal cholestasis—and particularly those who are suspected to have BA—undergo a diagnostic process developed in an interdisciplinary manner. Biliary atresia is confirmed or excluded by endoscopic retrograde cholangiography (ERC) and/or intraoperative visualization of the extrahepatic bile ducts [17]. Portoenterostomies were performed according to the original Kasai procedure in a standardized technique along with the following surgical key points: small transverse center right laparotomy, mobilization of the liver, intraoperative cholangiography (if necessary), preparation of the hepatoduodenal ligament and extensive dissection of the porta hepatis using magnification, preparing of a 40 to 50 cm Roux-Y-loop without valve creation, performing a funnel-shaped KPE, wedge liver biopsy, and no drainage. 

Postoperatively, the patients were monitored via intermediate care and oral feeding (with breast milk and/or a medium-chain triglycerides formula) was restarted within 24 h. The perioperatively given prophylaxis of antibiotics (trimethoprim) was continued for 10 to 14 days and then switched to oral for a minimum of six months. Fat-soluble vitamins and also ursodeoxycholic acid were prescribed with long-term intentions with respect to the clinical course. Post-Kasai, a budesonide 2 mg/dose rectal foam (Budenofalk™) was started on day five and with a continued daily application for three months. Patients of the control group were not treated by other steroids alternatively because we stopped anti-inflammatory adjuvant therapy since our high dosage study revealed no benefit in terms of jaundice free survival with native liver [18].

The follow-up of the patients was determined by the course of the disease. Babies with an uneventful development (colored stools, no jaundice, no ascites, decreasing bilirubin, age-appropriate weight gain) were initially followed up every three months, then with increasing intervals at 6 and 12 months, respectively. Patients with unfavorable outcomes were scheduled for LTx evaluation according to their individual course. Most of the patients were followed up in our own liver unit, although others did return to their local hospitals. Any healthcare colleagues involved were informed about the budesonide treatment and were asked to observe the patients thoroughly and to report any unexpected observations, particularly with regards to potential corticoid-induced symptoms.

### 2.2. Patients and Data Management

From the turn of the millennium, BA patients’ data were prospectively recorded in EBAR (European Biliary Atresia Registry) [18], which was incorporated in 2013 into the newly-established internet-based BARD-registry (Biliary Atresia and Related Diseases, www.BARD-online.com, accessed on 15 October 2021) [1]. The registry includes each patient’s initial entry dataset, a second data entry six months after the Kasai, then continued by infinite annual follow-ups or until the moment of LTx or death. Seamless data entry is supported by an e-mail reminder, which is automatically sent out to the user when follow-up data entries are due. The provider declares that the registry is in compliance with German data protection guidelines. Parents and patients of all BA patients since 2000 are informed that their pseudonymized data is used for scientific purposes exclusively and they can request the erasure of their data at any time without giving reasons.

Patients who changed to other centers were asked to agree with the same follow-up procedures. For these patients, data acquisition sometimes required active contact with the families, the pediatricians, and the liver units or transplant centers, respectively. In cases where the follow-up data was not completely recorded to the BARD-registry, traveling to participating centers for on-site data collection was necessary (D.F.).

The study group (SG) was built from patients born between February 2011 and October 2019, consisting of those whose parents agreed with an adjuvant budesonide treatment as described above. The controls were retrospectively recruited from our own patient cohort, born between March 2002 and October 2019, including those patients whose parents refused the budesonide treatment. Patients in the control group (CG) were also documented in the BARD-registry and the seamless follow-up procedure was the same as for the SG. The criteria for the case-matched CG were sex and age at the time of the KPW. Additionally, the following variables were included: gestational age, syndromic vs. non-syndromic form as well as bilirubin (total), AST, ALT, GGT, and liver fibrosis (calculated according to the Ishak classification) at the time of KPE. Cases that had been involved in other studies [17,19,20] were excluded.

As an outcome measurement, our definitions included the survival over all (SOA), survival with native liver (SNL) and jaundice-free survival with native liver, bilirubin < 20 µmol/L (jfSNL), determined at six months, two years after the KPE and October, 2021 (when the observation period was closed).

Neither of the registries routinely collect the parameters for steroid side effects. Therefore, we reviewed the files of those 82 patients separately, which are followed up in our pediatric liver unit for certain key parameters: percentile-related physical growth, the appearance of Cushing’s symptoms, elevated blood glucose, infections that require antibiotic treatment, and skeleton conspicuities like reduced bone mineral density (when x-rays were indicated).

### 2.3. Statistics

We analyzed quantitative data using SPSS V.24.0, considering results statistically significant when *p* < 0.05. Differences in survival rates were analyzed using Kaplan–Meier survival curves and significance was determined by log-rank test. To describe factors affecting survival, we used descriptive statistics, χ^2^ tests and multivariate analysis of variance.

## 3. Results

As of the reporting date of this still ongoing observational study, 95 BA patients born between 2011 and 2019 could be included (SG) while 81 patients born between 2002 and 2019 were matched for the CG. Twenty-two newborns, which have also been operated by the Kasai procedure during this period could not be considered because their parents did not agree to the experimental treatment with budesonide. However, they could only partially be included into the control group because only eight of them met the matching criteria. The follow-up of all 176 patients was truncated by the end of October 2021 with 100% completeness and no drop-off.

Concerning the inclusion criteria, no considerable difference could be found between both groups. A mild predominance was given for females (SG 62%/38% and CG 58%/42%), while the average age at KPE was 60 days (range 26–142) in the SG and 65 days (range 16–150) in the CG. None of the parameters were statistically different between both groups, except for preoperative bilirubin (SG: 136 µmol; CG 156 µmol). On the other hand, two parameters had a predictive value for all patients in terms of SOA: the survival with native liver was worse in patients with the syndromic form of BA, but the jfSNL was higher in those patients who had been operated on between days 31 and 60 (Appendix A). In addition, the following study group characteristics had been gathered: seven patients had congenital heart defects (e.g., hemodynamically significant ASD in five patients, tetralogy of Fallot, and hypoplastic left heart syndrome with a congenitally absent inferior vena cava in one patient each). Also, five patients had been tested positive for CMV. None of these parameters gave ground for exclusion – neither with respect to the budesonide treatment nor to the cause of death post-Kasai.

Survival with native liver at the defined reference dates was the main outcome objective. Six months after KPE, 78% of the SG survived without LTx as opposed to 73% of the controls. This difference becomes statistically significant after two years when 54% of the patients with budesonide live with their own liver in contrast to 32% of the control group—with the same being true for the jfSNL (*p* < 0.001). Our results, therefore, show a difference for six months after KPE as 55% in the SG vs. 35% in the CG, and 45% vs. 28% two years later (Table 1).

Overlooking the whole observation period of 20 years, the SOA of all patients was 89% and no difference could be found between both groups at any of the target points (Appendix A). In both groups, three patients died before the first measurement point at six months and in period up to two years after KPE, another five patients deceased in the SG, and nine in the CG. In the latter group another one died on the waiting list. These patients were not excluded from any statistical analysis. However, long-term survival with their native liver (130 months) was found to be 49% in the SG, with 48% even being jaundice free. In the control group, the SNL dropped after 229 months to 20% and to 18% for jfSNL, respectively, as shown in Figure 1.

At the end of the observation period, 12 additional BA patients with budesonide therapy, who already passed the six-month evaluation date after KPE, showed identical outcome in terms of SNL and jfSnL. 

With regard to steroid side effects, none of the 72 patients from the study group followed-up by our clinic showed a Cushing-like appearance or had elevated blood sugar levels. The physical development was reviewed only in those patients, who survived with their native liver for at least two years. In eight of these patients, the bodyweight six months after KPE was documented as either at or below the third percentile. All but one of them had a catch-up to the 15th or 50th percentile. Six months after KPE, six patients already presented as overweight at the 85th percentile, which did not decrease within the following 18 months.

## 4. Discussion

The first clinical corticosteroid trials after Kasai procedures date from the 1980s and are followed by numerous other studies and meta-analyses [10]. Consistently, they conclude that no benefit has been shown so far in terms of mid- or long-term improved survival with native liver. In this context, a recently published Cochrane review also stated that “further randomized, placebo-controlled trials are required to be able to determine if glucocorticosteroids may be of benefit in the postoperative management of infants with biliary atresia treated with Kasai portoenterostomy” [21,22].

Our series doesn’t fulfill these requirements but reveals for the first time that, according to long-term evaluation, BA patients with adjuvant therapy survive significantly longer with their native liver. Nevertheless, our clinical trial needs to be critically reviewed and inevitably requires a check for whether the study group is representative of all biliary atresia patients. In accordance with the literature, the patients in our SG are predominantly female and the syndromic form of BA was documented for 11% of patients, while other associated diseases, like congenital heart defects, were diagnosed in 7%. In addition, a 5% CMV positivity corresponds with other series. Summarizing the SG parameters and the control inclusion criteria, this study’s patients could be considered as representative of BA.

Another crucial factor concerns the outcome objectives of adjuvant therapy series in BA in general. Indeed, one may ask why the 2-year SNL and jfSNL of our study group is higher than in our controls but not different from other series reported by renowned liver units. This alleged inconsistency is explainable by the following arguments. Firstly, studies from East Asian centers cannot be directly compared to Caucasian series because the incidence of BA is different in both regions and a potentially diverging pathomechanism of postoperatively developing liver fibrosis is a matter for discussion. Secondly, BA studies in general use diverging outcome measures in terms of follow-up periods and nonuniform definition of jaundice-free SNL [23]. Finally, no other study provides a 100% follow-up of BA patients over a period of nearly 10 years in the SG and nearly 19 years in the CG (Appendix A). Herein, we explicitly refer to long-term survival, shown in Figure 1, where the jfSNL rate in the SG remains stable beyond 65 months. In the control group, the number of SNL patients drops continuously and falls below the 20% line, which is slightly lower than recorded in other long-term reports. However, particularly for this period, the indication and optimal timing of LTx are not regulated and depend upon many factors, which limits the comparability of long-term series. In principle, therefore, a strict one-to-one comparison of BA studies should be handled with great care and caution.

Besides the considerations of the comparability and reliability of diverging BA studies, two key aspects also need to be addressed. 

The first aspect concerns the question of why our regime of rectally-applied budesonide appeared to be more efficient than high dose prednisolone. The advantages of budesonide over commonly-used steroids like prednisolone might result from the approximately 15 times higher affinity to the steroid receptor and the 90% hepatic first-pass effect. The dose of rectal budesonide given (2 mg) to our patients is roughly equivalent to a dose of 10 mg/kg/d of prednisolone. This is comparable to high dose regimens of glucocorticoids used after Kasai portoenterostomy in former studies (Appendix A). Both characteristics allow a higher topical steroid concentration with less systemic steroid specific side effects (SSSE). In combination with the longer treatment, these factors might contribute to budesonide´s higher efficacy.

Budesonide was administered orally in AIH and PBC studies [11,12,13,14,15,16] leading to a complete portal venous drainage with topic effect on the liver. A similar effect was observed in adult patients with primary biliary cholangitis, whose biochemical disease markers improved after oral treatment with budesonide [GH]. We chose the rectal application due to its ease of application and based on the assumption that a higher dosage of budesonide in the rectum and colon versus the upper gastrointestinal tract could be achieved. Additionally, recent studies demonstrate the role of gut microbiota in experimental [24] and clinical [25] BA, as well as in the regulation of liver immunity [26]. However, only a little is known about the role of the gut liver axis in biliary atresia [27].

Besides the strong activity on the classic glucocorticoid receptor, budesonide has also been shown to have stronger noncanonical effects compared to corticoids, such as predniso(lo)ne. This includes rapid nongenomic effects via a receptor located in the cell membrane, which has been shown to contribute significantly to its action [28,29]. Furthermore, glucocorticoids are able to bind to and activate other receptors, while budesonide has less cross-reactivity with the mineralocorticoid receptor. However, budesonide is also known to be an agonist on the pregnane x receptor—a potential target for the treatment of cholestatic liver diseases [30,31]. Research activity in this broad field is still very new and little is known yet about the effects of a noncanonical pathway in human hepatobiliary diseases, while there are reports from animal studies about the effects of this pathway on hepatocytes [32].

The second aspect concerns indications for the use of budesonide, which, in principle, does not differ between pediatric patients and adults. The leading diagnoses are eosinophilic esophagitis, autoimmune hepatitis, active ulcerative colitis, and a mild to moderate course of Crohn’s disease. The most frequently reported adverse events are aphthous stomatitis, acne vulgaris, moon face, headache after oral administration, burning pain in the rectum, anal fissure, frequent urge to defecate, and bleeding after rectal use. These side effects depend on the steroid dosage and duration, while reduced glucose tolerance, growth retardation, increased appetite following weight gain, increased risk of infection and osteopenia can occur with long-term administration [33]. The critical aspect of this study is that the indication, dosage, and duration of any budesonide application are not given to neonates with BA. For this reason, we observed the study group patients with a particular emphasis on steroid-related side effects, which usually have no part in any BA follow-up protocol. Only eight out of 72 patients were documented as having growth retardation that had finally caught up, which could be considered as a steroid-related side effect. However, a more meticulous follow-up protocol is mandatory when budesonide is used routinely in BA patients.

## 5. Conclusions and Perspectives

In the history of biliary atresia treatment, there are two milestones to be highlighted: firstly, Kasai portoenterostomy and, secondly, liver transplantation achievements, namely, split liver Tx and living-related Tx. Apart from these two surgical procedures, no progress has been made in decades, particularly regarding the different protocols of adjuvant therapy post-KPE. The majority of studies report on corticosteroids in various dosages but all of them have failed to identify evidence in terms of longer survival with the patients’ own livers. New studies with antiviral therapy, the administration of immunoglobulins, N-acetylcysteine, intestinal bile salt transport inhibitors, obeticholic acid [8], the Chinese herbs mixture “Inchinko-to” [34] and other treatments are on the way. Currently, however, our work on rectally-administered budesonide after KPE is the only existing study demonstrating a significant decrease of the need for liver transplantation in BA in respect of both mid- and long-term objectives. In light of these results, we advocate for a prospective observational multicenter study with a long follow-up and a major focus on steroid-related side effects, as well as clinical and basic research on the gut–liver axis factors (41) in biliary atresia.

## Figures and Tables

**Figure 1 jcm-10-05758-f001:**
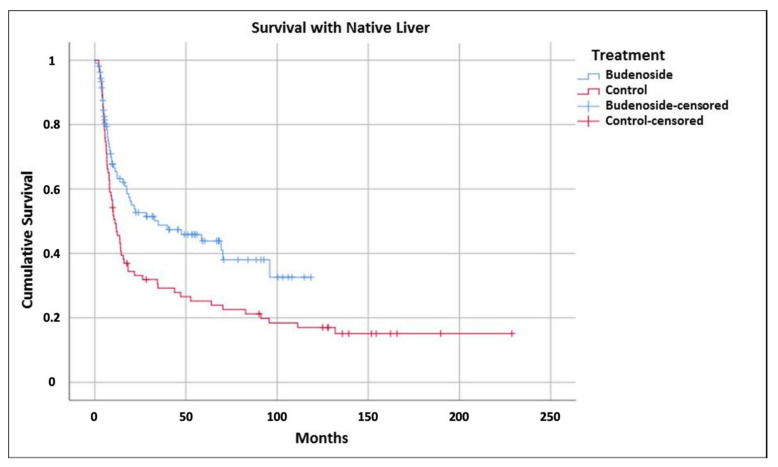
After two years, survival with native liver was 54% in the study and 32% in the control group (*p* < 0.001).

**Table 1 jcm-10-05758-t001:** Outcome after Kasai procedure with and without adjuvant budesonide therapy at six months and two years after the Kasai portoenterostomy (KPE).

	Study/ Control Group	SOA	*p*-Value n.s.	SNL	*p*-Value n.s.	jfSNL	*p*-Value
6 months post KPE N = 176	SG	99% (94/95)	n.s.	78% (74/95)	n.s.	55% (52/95)	n.s.
	CG	100% (81/81)		73% (60/81)		35% (29/81)	
2 years post KPE N = 176	SG	92% (87/95)	n.s.	54% (51/95)	*p* < 0.001	45% (43/95)	*p* < 0.001
	CG	88% (73/81)		32% (26/81)		28% (23/81)	

Study group (SG) and control group (CG), survival over all (SOA), survival with native liver (SNL) and jaundice-free survival with native liver (jfSNL). n.s., nonsignificant.

## Data Availability

Not applicable.

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
