# Peer review of "Adjuvant Therapy with Budesonide Post-Kasai Reduces the Need for Liver Transplantation in Biliary Atresia"

_jcm, 2021, doi:10.3390/jcm10245758_

Round 1

Reviewer 1 Report

Inflammation is thought to be a key factor in the pathogenesis of many cases of biliary atresia (BA), while prednisolone is the most frequently prescribed steroid in most studies.

The Authors from Hannover presented interesting results on the usage of rectally-administered budesonide as adjuvant therapy after Kasai hepatoportoenterostomy. They recruited 95 BA patients who received budesonide 2 mg/dose rectal foam daily for three months. The primary endpoint was survival with native liver (SNL) which was determined at six months and two years after the Kasai procedure. SNL was statistically different after two years. No steroid-related side effects have been observed.

This is the first published study on the usage of budesonide after Kasai treatment which demonstrated a significantly longer survival with native liver in patients with BA after adjuvant therapy. 

Author Response

Dear Editors    

Dear Reviewers

Thank you for positively evaluating our study and for the useful comments. We answer to each question point-by-point:

Reviewer #2 wrote that “the following concerns should be addressed”:

  1. My first concern is the patients in this study. There is a description of inclusion and exclusion criteria in the method, but it should be clearly stated how many cases existed during the period and how many of them were excluded. Moreover, there seem to be some discrepancies in the number of patients in Table 1. It is necessary to state the numbers more precisely.

During the period between 2011 and 2019, we operated on 117 BA-patients over all. Ninety-five of them received adjuvant therapy according to our budenoside protocol, while the parents of the remaining 22 newborns didn´t agree. These patients were not treated by other steroids alternatively because we stopped anti-inflammatory adjuvant therapy since our high dosage study (Ref. 18) revealed no benefit in terms of jaundice-free survival with native liver. However, these 22 patients could only partially be included in the control group because only eight of them met the matching criteria. (line 119 ff, line 176 ff, line 201)

Concerning Table 1, we thank the reviewer for carefully reviewing the text, tables, and all other data. We, indeed, mixed the total number of patients up with data from another study. The table is now correct.

  1. The second concern is the status of steroids usage in CG. If there were no patients with steroids, such as prednisolone, it is better to clearly state that point and why those patients didn’t use steroids as adjuvant therapy.

We thank the reviewer for this comment and we´ve added this information (see above ) to the text.

  1. The third concern is the basis of budesonide dosage as an adjuvant therapy with Kasai operation. In the discussion section, there is the following description,” The dose of rectal budesonide given (2 mg) to our patients is roughly equivalent to a dose of 10 mg/kg/d of prednisolone.”. However, there was no reference in this description.

We thank the reviewer for this query, too. However, we can´t provide a particular reverence here, because we just asked the department of pharmacology to provide the calculation of an equivalent dosage.

  1. The fourth concern is steroid side effects. In the results section, there is the following description,” With regard to steroid side effects, none of the 72 patients from the study group followed-up by our clinic”. In this context, does that mean that the authors couldn’t investigate the side effect of budesonide in the whole SG case?

The assumption of the reviewer that the observation of budenoside side effects of all 95 SG patients is obviously incomplete, is correct. The reason is that 22 BA patients, which were operated on by us, were not followed up completely by ourselves. These patients returned to the referring hospital or pediatrician for follow-up (some of which came from other countries). In as much as we evaluated the follow up retrospectively, we could only refer to the primarily disease-related outcome parameters, which were available in all patients (SOA, SNL, jfSNL, bilirubin at 6 months, 1 and 2 years after KPE and at the end of the follow-up period). However, monitoring of additional clinical parameters was not routinely done in these 22 patients and could, therefore, not be used in this particular matter.  

Thank you for your consideration,

yours sincerely

Claus Petersen

Reviewer 2 Report

Thank you for the opportunity to review this article.

In this study, the authors investigated the efficacy of adjuvant therapy with budesonide. This subject would certainly be of interest to pediatric hepatologists and pediatric surgeons.

However, the following concerns should be addressed:

  1. My first concern is the patients in this study. There is a description of inclusion and exclusion criteria in the method, but it should be clearly stated how many cases existed during the period and how many of them were excluded. Moreover, there seem to be some discrepancies in the number of patients in Table 1. It is necessary to state the numbers more precisely.
  2. The second concern is the status of steroids usage in CG. If there were no patients with steroids, such as prednisolone, it is better to clearly state that point and why those patients didn't use steroids as adjuvant therapy.
  3. The third concern is the basis of budesonide dosage as an adjuvant therapy with Kasai operation. In the discussion section, there is the following description," The dose of rectal budesonide given (2 mg) to our patients is roughly equivalent to a dose of 10 mg/kg/d of prednisolone.". However, there was no reference in this description.
  4. The fourth concern is steroid side effects. In the results section, there is the following description," With regard to steroid side effects, none of the 72 patients from the study group followed-up by our clinic". In this context, does that mean that the authors couldn't investigate the side effect of budesonide in the whole SG case?

Author Response

Dear Editors    

Dear Reviewers

Thank you for positively evaluating our study and for the useful comments. We answer each question point-by-point:

Reviewer #2 wrote that “the following concerns should be addressed”:

  1. My first concern is the patients in this study. There is a description of inclusion and exclusion criteria in the method, but it should be clearly stated how many cases existed during the period and how many of them were excluded. Moreover, there seem to be some discrepancies in the number of patients in Table 1. It is necessary to state the numbers more precisely.

During the period between 2011 and 2019, we operated on 117 BA-patients over all. Ninety-five of them received adjuvant therapy according to our budenoside protocol, while the parents of the remaining 22 newborns didn´t agree. These patients were not treated by other steroids alternatively because we stopped anti-inflammatory adjuvant therapy since our high dosage study (Ref. 18) revealed no benefit in terms of jaundice-free survival with native liver. However, these 22 patients could only partially be included in the control group because only eight of them met the matching criteria. (line 119 ff, line 176 ff, line 201)

Concerning Table 1, we thank the reviewer for carefully reviewing the text, tables, and all other data. We, indeed, mixed the total number of patients up with data from another study. The table is now correct.

  1. The second concern is the status of steroids usage in CG. If there were no patients with steroids, such as prednisolone, it is better to clearly state that point and why those patients didn’t use steroids as adjuvant therapy.

We thank the reviewer for this comment and we´ve added this information (see above ) to the text.

  1. The third concern is the basis of budesonide dosage as an adjuvant therapy with Kasai operation. In the discussion section, there is the following description,” The dose of rectal budesonide given (2 mg) to our patients is roughly equivalent to a dose of 10 mg/kg/d of prednisolone.”. However, there was no reference in this description.

We thank the reviewer for this query, too. However, we can´t provide a particular reverence here, because we just asked the department of pharmacology to provide the calculation of an equivalent dosage.

  1. The fourth concern is steroid side effects. In the results section, there is the following description,” With regard to steroid side effects, none of the 72 patients from the study group followed-up by our clinic”. In this context, does that mean that the authors couldn’t investigate the side effect of budesonide in the whole SG case?

The assumption of the reviewer that the observation of budenoside side effects of all 95 SG patients is obviously incomplete, is correct. The reason is that 22 BA patients, which were operated on by us, were not followed up completely by ourselves. These patients returned to the referring hospital or pediatrician for follow-up (some of which came from other countries). In as much as we evaluated the follow up retrospectively, we could only refer to the primarily disease-related outcome parameters, which were available in all patients (SOA, SNL, jfSNL, bilirubin at 6 months, 1 and 2 years after KPE and at the end of the follow-up period). However, monitoring of additional clinical parameters was not routinely done in these 22 patients and could, therefore, not be used in this particular matter.  

Thank you for your consideration,

yours sincerely

Claus Petersen

Round 2

Reviewer 2 Report

Thank you for your sincere response to my comments.
I have no further concern.